# Feasibility of Near-Infrared Spectroscopy for Identification of L-Fucose and L-Proline—Towards Detecting Cancer Biomarkers from Saliva

**Miia O. Hurskainen** [1,2,*] , **Jaakko K. Sarin** [2,3,4] , **Sami Myllymaa** [1,2,3] , **Wilfredo A. González-Arriagada** [5] , **Arja Kullaa** [6] and **Reijo Lappalainen** [1,2]

1 SIB Labs, University of Eastern Finland, FI-70211 Kuopio, Finland; sami.myllymaa@uef.fi (S.M.); reijo.lappalainen@uef.fi (R.L.)
2 Department of Applied Physics, University of Eastern Finland, FI-70211 Kuopio, Finland; jaakko.sarin@uef.fi
3 Diagnostic Imaging Center, Kuopio University Hospital, FI-70211 Kuopio, Finland
4 Department of Medical Physics, Tampere University Hospital, FI-33521 Tampere, Finland
5 Centro de Investigación e Innovación Biomédica, Facultad de Odontología, Universidad de los Andes, Las Condes 7550000, Chile; wgonzalez@uandes.cl
6 Institute of Dentistry, School of Medicine, University of Eastern Finland, FI-70211 Kuopio, Finland; arja.kullaa@uef.fi
* Correspondence: miia.hurskainen@uef.fi

**Abstract:** Near-infrared spectroscopy (NIRS) is a non-ionizing optical technique that can be used to quantify proteins, carbohydrates, fats, and other organic and biological substances. The aim of this study was to determine the ability of NIRS to identify different concentrations of L-fucose and L-proline solutions by utilizing different NIR spectral regions. NIR spectra of solid L-fucose and L-proline, their aqueous solutions in different concentrations, and the spectra of saliva samples collected from two patients with oral squamous cell carcinoma (OSCC) were studied. Differences in spectra of the pure solid reference samples and water were most noticeable in spectral regions 800–1250 nm and 1418–1867 nm. The saliva sample with an atypically high concentration of oral cancer biomarkers showed a similar spectral feature between 1530–1650 nm as the liquid samples with cancer biomarkers. In addition, a fine k-nearest neighbors (kNN) classifier was trained to differentiate the aqueous solutions and achieved 75.97% validation accuracy. The preliminary study presents that NIRS can be utilized to detect differences in spectra between the different biomarker concentrations in aqueous solutions. However, the qualitative measures may have resulted in limited sensitivity, which could be enhanced by additional samples and using a measurement probe dedicated to fluid measurements.

**Keywords:** near-infrared spectroscopy; L-fucose; L-proline; saliva; oral cancer

## 1. Introduction

Oral diseases threaten overall wellness and increase the risk for several systemic diseases. The problem with diagnosing oral diseases is their overlapping symptoms. Visual observation and palpation used in clinical practices lead to poor accuracy of diagnosis. Oral potential malignant disorders (OPMD) and oral squamous cell cancers (OSCC) are diagnosed using histopathological analysis of surgically removed tissue biopsies stained using hematoxylin and eosin dyes [1,2], which are highly invasive, expensive, and time-consuming [3].

Optical methods based on fluorescence or chemiluminescence [1,2] can be used to confirm the presence of lesions but these methods cannot differentiate between low-risk and high-risk lesions [4,5]. Spectral cameras, such as VELScope, have not been able to distinguish between pre-malignant and malignant tissues more reliably than experienced

clinicians [4,6–10]. Bioimpedance spectroscopy (BIS) can detect oral lesions, but the diversity of probe designs makes it difficult to draw reliable conclusions on its diagnostic potential [5]. Genetic analysis can help in the selection of potential locations for biopsies, and add value to histopathological diagnosis [11]. Optical coherence tomography (OCT) in the detection of OPMDs and OSCCs has been studied by Jerjes et al. [12]. They concluded that the ability of the OCT to distinguish different oral lesions is poor. Liao et al. [13] concluded that fluorodeoxyglucose (FDG) positron emission tomography (PET) is valuable in planning of the patient's treatment strategy.

In biomedical applications, near-infrared spectroscopy (NIRS) can be used in vivo to distinguish healthy tissues from pre-malignant and malignant tissues and aid the diagnostics due to its excellent penetration depth and its minimal requirement for sample preparation [14–22]. Furthermore, optical spectroscopy can be performed without sample extraction, enabling the non-invasive assessment of tissue [23–27]. For example, spectroscopic techniques are used to assess periodontal inflammation [28], and various cancers, including skin [15–19,29], breast [30,31], oral [32,33], oropharyngeal, and laryngeal cancer [23].

*Near-Infrared Spectroscopy in Assessing Salivary Biomarkers*

NIRS could be potentially used as a non-invasive intra-oral analysis technique. In vivo studies of the oral cavity should account for the presence of saliva and consider it as diagnostic fluid complementing the information gathered from the oral mucosa [34–39]. The use of saliva instead of blood or serum in diagnosing various conditions, such as cancers [30,31,35,40,41], diabetes [20], oral leukoplakia [35,40], Sjögren's syndrome [36], or testing for COVID-19 [42–44], is a growing area of research. Saliva consists of 94–99% of water [34,42] but it also contains biomarkers, including locally expressed proteins and substances, such as DNA, RNA, mucins, amino acids, enzymes, and primary metabolites. The concentrations of saliva biomarkers change with different diseases [20,35,45–47]. Compared to blood and serum, saliva has a less complex and varying composition [20,35]

Oral diseases cause abnormal changes in oral mucosal tissue. Oral epithelial cells constantly shed into the saliva [35,48], and oral cancer cells start to clear into the saliva at the early stages of cancer [35,48]. Saliva can exchange substances with systemic circulation via diffusion or active transport due to its direct contact with oral mucosa [35,36,48,49]. Collecting saliva samples is easier (e.g., feasible at home), less expensive, less uncomfortable for the patients, and safer (excludes the risk of needle stick) than collecting blood samples or biopsies, thereby allowing the increased use of saliva as diagnostic fluid in the future, especially for a screening of potential diseases. However, to date, the biomedical studies of NIRS have mainly focused on studying skin [15–18,29] or cartilage [50–52]. Murayama et al. [38] utilized NIRS to study saliva of oral cancer patients and healthy controls with the capillary tube Fourier transformation–NIR (FT–NIR) [53]. However, no biomarkers were investigated in their study. Other NIR studies of saliva focus on hemodynamic signals [54,55] or forensic applications [56]. However, the previous studies do not utilize NIRS to characterize saliva with the known oral cancer biomarkers.

A previous nuclear magnetic resonance (NMR) study by Mikkonen et al. [45] studied potential salivary biomarkers of oral cancer. Their study showed that oral cancer patients exhibited significantly increased concentrations of salivary L-fucose while salivary L-proline levels were significantly decreased compared to healthy controls [36,45]. However, saliva has high water content [34–36,45], and water is a strong NIR absorber. In this proof-of-concept study, we aimed to identify the spectral regions for discriminating solutions with different concentrations of L-fucose and L-proline and water. These regions could be further utilized in salivary diagnostics to recognize and discriminate between healthy, patients suffering from premalignant conditions, and cancerous patients based on saliva samples.

## 2. Materials and Methods

### 2.1. Materials

We studied NIR spectra of salivary biomarkers: L-fucose [36,45,46,57,58] (L-(-)-Fucose ≥ 99%, Sigma Aldrich, Saint Louis, MO, USA) and L-proline [36,45] (L-Proline-Reagent Plus* ≥ 99%, Sigma Aldrich, Saint Louis, MO, USA), diluted into distilled water. Both substances have a solubility of 50 mg/mL in water. The measurements were repeated in three independent sequences. For each sequence, four solutions were prepared for both biomarkers with a serial dilution protocol as follows: a mass equivalent to the solubility of the substance, 50 mg/mL for both L-fucose and L-proline, was weighed, and then mixed with 1 mL of Millipore–water (18.2 MΩ). This means that the solutions with the highest concentrations have 50 mg of substance in 1 mL of water. The concentrations of solutions in mg/mL and μM are presented in Table 1. The aqueous solutions F1 to F3 and P1 to P3 have a higher concentration of L-fucose and L-proline than human saliva while the F4 and P4 solutions have a concentration similar to that of human saliva [45,46]. Mikkonen et al. [45] determined the concentrations of L-fucose and L-proline in the saliva of healthy controls (n = 30). For L-fucose, the concentrations were 189.1 (100.6–284.7) μM which is about 0.031 (0.016–0.047) mg/mL. For L-proline, the concentrations were 610.1 (318.5–1244.3) μM which is about 0.070 (0.037–0.143) mg/mL. In the study of Sharma et al. [46], salivary L-fucose in OPMDs and oral cancer were determined to be 7.02 mg/dl (0.0702 mg/mL) and 11.66 mg/dL (0.1166 mg/mL), respectively, while saliva from healthy controls had only 3.18 mg/dL (0.0318 mg/mL) of L-fucose. No similar study determining the concentration of L-proline in saliva was found. The concentrations of salivary metabolites identified from patients suffering OSCC [45] and their healthy controls [45] are listed in Table 2.

Reference samples were prepared by pressing pure L-fucose and L-proline powders into 3 mm thick tablets. These reference samples were used as no previous measurements with L-fucose, or L-proline had been performed with the NIRS measurement setup used in this study. The measurements were performed using three tablets of each substance by repeating the measurements six times on each tablet.

In this preliminary study, saliva samples of two OSCC [45,59] patients, a 73-year-old male patient (sample 1) and a 76-year-old male patient (sample 2), were used. Both samples were collected in Brazil (Piracicaba Dental School, University of Campinas). The samples were collected by González-Arriagada et al. [59] and further analyzed by Mikkonen et al. [45]. Patients not diagnosed with OSCC were not included. The age range of the patients in the study by González-Arriagada et al. [59] was 52–76 years (mean age 61.6 ± 9.6 years). The range for healthy controls was 42–74 years (mean age 54.4 ± 9.0 years). The saliva samples were collected unstimulated, meaning that the patients passively drooled the saliva into a sterile glass cup for 5 min [45]. The clinical characteristics of the patients are described in a study by Mikkonen et al. [45]. Saliva sample 1 has high L-fucose and L-proline concentrations 694 μM (0.106 mg/mL) and 799 μM (0.092 mg/mL), respectively. Saliva sample 2 has low L-fucose and L-proline concentrations 67 μM (0.011 mg/mL) and 157 μM (0.018 mg/mL), respectively. The collection of saliva and its use in the study was approved by the Ethics Committee for Human Studies, Piracicaba Dental School, Brazil (protocol number 142/2010), and written consent was obtained from every participant.

**Table 1.** The concentrations of L-fucose and L-proline solutions.

| Solution | mg/mL | μM | Marked in Text |
|---|---|---|---|
| L-Fucose 1 | 50 | $3.05 \times 10^5$ | F1 |
| L-Fucose 2 | 5 | $3.05 \times 10^4$ | F2 |
| L-Fucose 3 | 0.5 | $3.05 \times 10^3$ | F3 |
| L-Fucose 4 | 0.05 | $3.05 \times 10^2$ | F4 |
| L-Proline 1 | 50 | $4.34 \times 10^5$ | P1 |
| L-Proline 2 | 5 | $4.34 \times 10^4$ | P2 |
| L-Proline 3 | 0.5 | $4.34 \times 10^3$ | P3 |
| L-Proline 4 | 0.05 | $4.34 \times 10^2$ | P4 |

**Table 2.** Concentrations of salivary metabolites of oral squamous cell cancer (OSCC) (n = 8) patients and their healthy controls (n = 30) identified in an NMR study by Mikkonen et al. [45].

| Metabolite | OSCC [µM] | Controls [µM] |
|---|---|---|
| Acetate | 2559.8−9344.8 | 1977.7−5239.5 |
| Alanine | 47.1−515.9 | 53.0−173.0 |
| Butanol | 17.2−190.5 | 16.8−84.3 |
| Butyrate | 33.9−266.4 | 25.9−128.4 |
| Choline | 12.2−43.7 | 14.2−24.7 |
| Formate | 191.4−426.2 | 77.0−433.3 |
| L-fucose | 302.0−1527.2 | 100.6−284.7 |
| Glycine | 103.2−719.1 | 241.1−923.6 |
| Lactate | 71.5−1132.9 | 140.4−324.6 |
| Methanol | 36.6−208.1 | 51.4−121.5 |
| Methylamine | 1.7−66.5 | 1.9−5.7 |
| Phenylalanine | 41.9−147.6 | 59.1−123.6 |
| L-proline | 104.1−799.9 | 318.5−1244.3 |
| Propionate | 319.9−2157.6 | 251.1−1028.4 |
| Pyruvic acid | 12.6−73.1 | 7.2−33.3 |
| Succinate | 24.5−214.3 | 47.1−71.9 |
| Taurine | 72.1−195.4 | 104.7−205.1 |
| Tyrosine | 42.3−173.5 | 55.3−165.5 |
| 1,2-propanediol | 32.7−2465.4 | 21.7−54.1 |

## 2.2. Near Infrared Spectroscopy

The setup used in this study consisted of two spectrometers (AvaSpec-ULS2048L and AvaSpec-NIR256-2.5-HSC, Avantes BV, Apeldoorn, The Netherlands), a light source (AvaLight-HAL-(S)-Mini, Avantes BV), and a customized arthroscopic optical fiber probe (Avantes BV, Apeldoorn, The Netherlands). A similar set-up was used to study articular cartilage and subchondral bone by Sarin et al. [60]. The optical fiber probe is not initially designed for liquid measurements. For the AvaSpec-ULS2048L, the integration time was 1.5 ms for pure L-fucose, aqueous L-fucose, and L-proline solutions and saliva sample 2. For pure L-proline and saliva sample 1, the reference spectrum was saturated with the integration time of 1.5 ms; thus, the time was adjusted to 1.05 ms. The acquired spectral region was 319–1093 nm, the resolution of the spectrometer was 0.6 nm, and the averaging number was 100 scans. For the AvaSpec-NIR256-2.5-HSC, the integration time was 30 ms, the averaging number was 100 scans, the acquired spectral region was 943–2503 nm, and the resolution was 6.4 nm. The measured data is provided as supplementary materials (Table S1: Data sheet). Sample drops were placed on a reflective standard (Spectralon, SRS-99, Labsphere Inc., North Sutton, NH, USA) on top of a sample holder (Figure 1b), which was controlled by the motion controller (Figure 1a). The reflective standard was cleaned with tissue paper after each measurement before pipetting another drop on top of the reflective standard, making the sample deposition simple, fast, and inexpensive. Spectral measurements were performed at the exact same coordinates (x, y, z) with each drop. Preliminary tests revealed that optimal signal-to-noise ratio (SNR) was achieved with the sample volume of 20 µL which was standardized for the actual measurements.

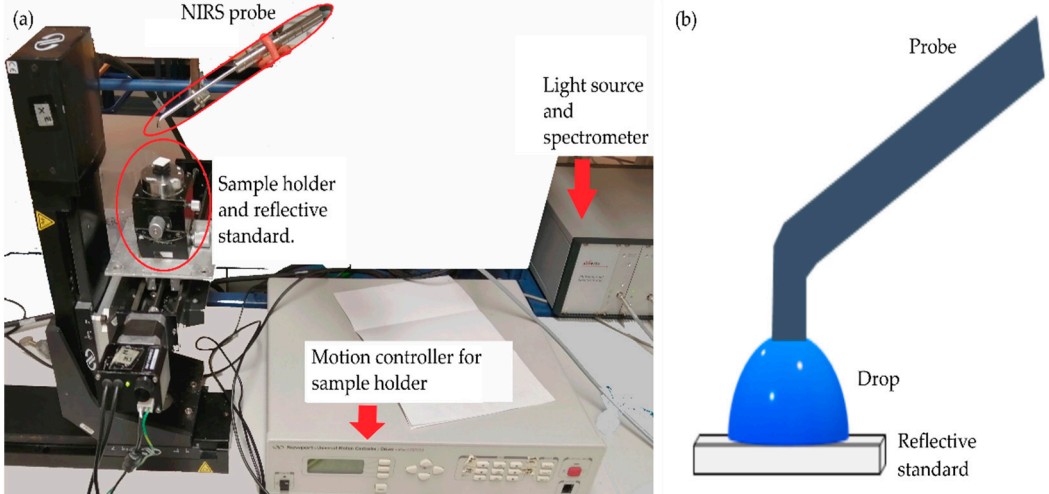

**Figure 1.** The NIRS set-up (**a**), and schematic presentation of the drop on top of reflective standard with the probe touching the drop (**b**).

### 2.3. Spectral Analysis

Before sample measurements, dark (non-reflective standard, 0.07% relative reflectance) and white references (reflective standard, Spectralon, SRS-99, Labsphere Inc., North Sutton, NH, USA, 99% relative reflectance) were measured within the wavelength region 320–2453 nm. The reflection from the drop surface can be neglected as the contact between the probe and sample drop was always ensured. Each measurement was repeated 6 times for each substance. A 20 μL drop of an aqueous solution was pipetted onto the reference material and the spectrum was measured. The measured data is provided as supplementary material (Table S1: Data sheet). The reflective material and the probe were then wiped clean, and the protocol was repeated. In total, 18 spectra for each concentration of aqueous L-fucose and L-proline solutions were acquired. An average of all 18 measurements was calculated from the raw data. Absorbance for transflection mode was calculated using the Beer–Lambert law (Equation (1))

$$A = -\log_{10}\frac{S-D}{W-D},\qquad(1)$$

where $S$ is the spectrum of the sample, $D$ is the dark reference spectrum, and $W$ is the white reference spectrum.

The values calculated using Equation (1) were normalized using standard normal variate (SNV). A Savitzky–Golay filter (degree = 3, window = 49 for 319–1100 nm, and degree = 3, window = 19 for 943–2500 nm) was used to smooth the data and calculate the 2nd derivative for each averaged spectrum (MATLAB R2019a, The MathWorks, Natick, MA, USA, 2019). Area normalization was used with the derivative spectra. The area normalization was calculated as (Equation (2))

$$S_{AN} = \frac{S}{\sum|S_p|},\qquad(2)$$

where $S_{AN}$ is area normalized spectra and $S_p$ is point of spectrum. The area normalization was done over 455–2370 nm. If solutions appeared in the order of F1, F2, F3, F4, and water, or P1, P2, P3, P4, and water, meaning an increase in water content, a black rectangle was used to mark those wavelengths. If the solutions appeared in the order of F1, F2, F3, and F4, or F1, F2, F3, and water, or P1, P2, P3, and P4, or P1, P2, P3, and water, assuming the F4 and P4 solutions and pure water having almost equal water content, these wavelengths were marked with a light green rectangle.

## *2.4. Multivariate Analysis*

A fine k-nearest neighbor (kNN) was applied to investigate if different concentrations of L-fucose and L-proline are distinguishable from each other and water. The confusion matrix of the cross-validated kNN from spectral regions 680–1400 nm and 1550–1850 nm was calculated. The spectral regions 1400–1550 nm and 1850–2500 nm had relatively poor SNR due to high water absorbance and, thus, these regions were excluded. Exclusion substantially increased the performance of the classifier. The spectral measurements with poor quality were manually excluded from the validation. The kNN model was trained individually to spectra without calculating the average of all spectra. The spectra were calculated using Equation (1), and the values were normalized using SNV. A Savitzky–Golay filter was used to smooth the data and calculate the 1st and 2nd derivatives for each spectrum (MATLAB R2019a, The MathWorks, Natick, 2019). Different window sizes for the Savitzky–Golay filter were tested for both the 1st and 2nd derivatives. The F1-score was calculated with confusionStats(group, grouhat) function by Cheong [61]. The input data was cross-validated (k-fold = 10) with a hundred repetitions.

## *2.5. Statistical Analysis*

One-way analysis of variance (ANOVA) was carried out to test the reproducibility of the sample deposition method. ANOVA was run to all spectra, that were included in the analysis, measured from one independent solution of each concentration for both L-fucose and L-proline. The statistical analysis was carried out with MATLAB (MATLAB R2019a, MathWorks Inc., Natick, MA, USA, 2019).

## 3. Results

Figure 2 presents the smoothed (Figure 2a), and SNV (Figure 2b) NIR spectra of pure L-fucose, L-proline, and water. Figure 3 presents the 2nd derivative (Figure 3a) NIR spectra of aqueous solutions of L-fucose and L-proline and water, and a close-up of the 2nd derivative NIR spectra from 880 to 889 nm (Figure 3b). Figures 4 and 5 present the 2nd derivative NIR spectra of aqueous solutions of L-proline and water, and saliva and water, respectively. The spectra of pure L-fucose and L-proline tablets are drastically different compared to that of water (Figure 2) as expected. The major differences are in the spectral regions 800–1250 nm and 1418–1867 nm. Spectral peaks of L-fucose and L-proline are in line with the current literature of their molecular groups [62–64] (Tables 3 and 4). Water has distinctive peaks at 970 nm, 1450 nm, and 1940 nm [62,65,66]. Two spectrometers were used in this study and the overlap of the spectra from both spectrometers is observed between 1000−1100 nm in Figures 2–5.

A systematic order, marked with both black and green, was observed with aqueous L-fucose solutions of different concentrations and pure water (Figure 3a) within the wavelengths of 695–969 nm, 718–719 nm, 869–888 nm, 907–908 nm, and 1013–1029 nm. The 718–719 nm, 869–888 nm, and 907–908 nm are assigned to the 3rd overtone CH stretching, and the 1011–1029 nm are assigned to the 2nd overtone OH stretching [62–64]. The same comparison shows that L-proline solutions differ from pure water (Figure 4) within 778–784 nm and 819–820 nm, that is assigned to the 3rd overtone NH stretching [62–64]. Wavelengths in the NIR spectra demonstrating a systematic order between aqueous biomarker solutions and pure water (Figures 3 and 4) are separately compared to the commonly known NIR signals of molecular groups [62–64] (Tables 3 and 4).

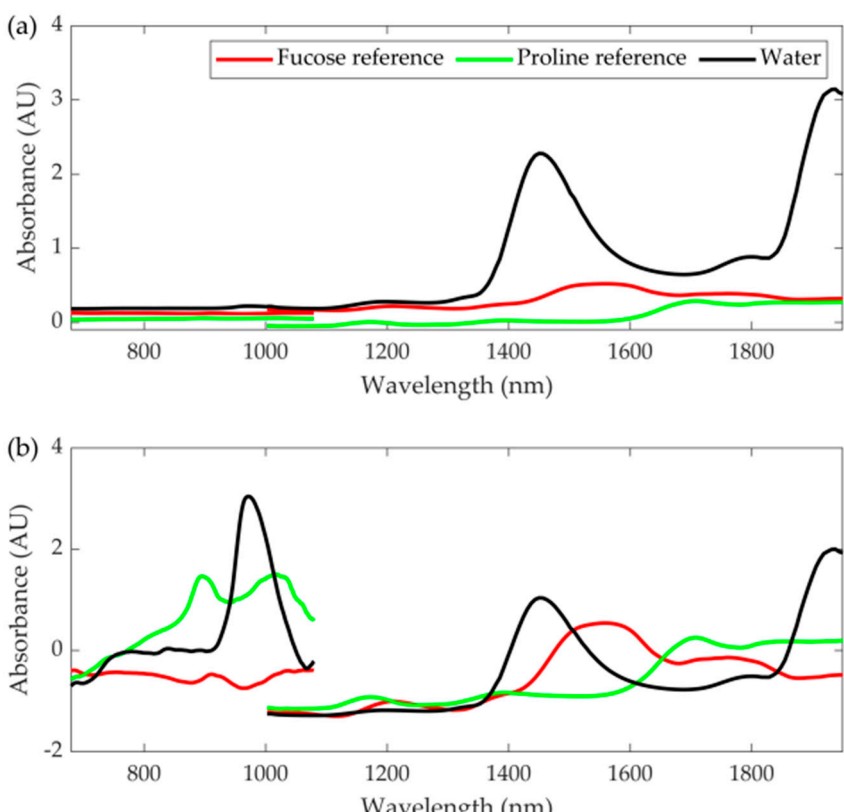

**Figure 2.** The smoothed (**a**) and normalized (**b**) NIR spectra of L-fucose tablet, L-proline tablet, and pure water.

**Table 3.** Wavelength areas of L-fucose solutions that were recognized to have a systematic order from F1 → F2 → F3 → F4 to water were marked black. The wavelength areas where the solutions were in order from F1 → F2 → F3 to F4 or F1 → F2 → F3 to water were marked green. The corresponding molecular groups are named on the right. Wavelengths under 680 nm belong to the UV-VIS region and are not of interest in this study.

| Wavelength Area (nm) Black | Wavelength Area (nm) Green | NIR Absorption Bands Molecular Groups [62–64] |
|---|---|---|
| 695–696 | 695–696 | |
| 718–719 | 718–719 | 3rd overtone CH stretching |
| | 786 | 3rd overtone CH stretching |
| 869–888 | 880–888 | 3rd overtone CH stretching |
| 1011–1029 | 1019–1029 | 2nd overtone OH stretching & water |

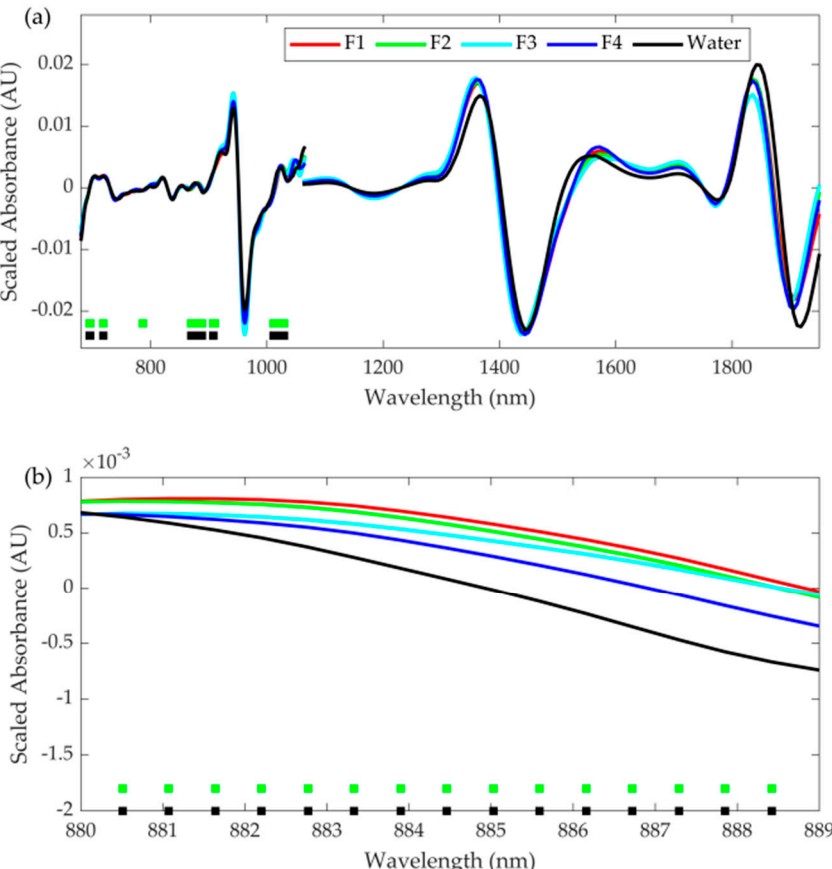

**Figure 3.** The 2nd derivative NIR spectra of aqueous L-fucose (**a**) and 2nd derivative spectra as a close-up from 880 to 889 nm (**b**). Each spectrum of each concentration shows the average of 3 independent solutions. The NIR spectrum of pure water is shown in comparison. The wavelength areas presenting a systematic order with the solutions of different concentrations and water (F1 → F2 → F3 → F4 → water) are marked with a black rectangle, whereas solutions appearing in the order F1 → F2 → F3 to F4 or F1 → F2 → F3 to water were marked with a light green rectangle.

**Table 4.** Wavelength areas of L-proline solutions that were recognized to have a systematic order Figure 1. → P2 → P3 →P4 to water were marked black. The wavelength areas where the solutions were in order from P1 → P2 → P3 to P4 or P1 → P2 → P3 to water were marked green. The corresponding molecular groups are named on the right. Wavelengths under 680 nm belong to the UV-VIS region and are not of interest in this study.

| Wavelength Area (nm) Black | Wavelength Area (nm) Green | NIR Absorption Bands Molecular Groups [62–64] |
|---|---|---|
| 725 | 725 | 4th overtone CH stretching |
| 781–784 | 778–796 | 3rd overtone NH stretching |
| 819–820 | 819–820 | 3rd overtone NH stretching |
| 850–864 | 850–864 | 3rd overtone CH stretching |
| | 1050 | 2nd overtone N–H stretching, 2nd overtone O–H stretching |
| 1906 | 1906 | |

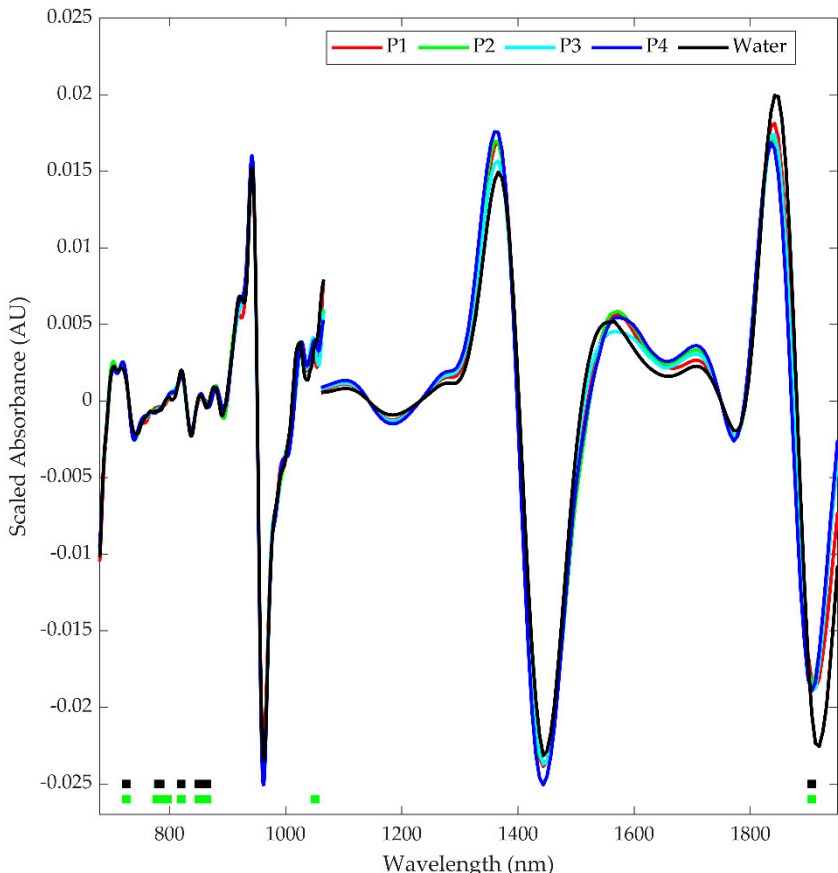

**Figure 4.** The 2nd derivative NIR spectra of aqueous L-proline. Each spectrum of each concentration shows the average of 3 independent solutions. The NIR spectrum of pure water is shown in comparison. The wavelength areas presenting a systematic order with the solutions of different concentrations and water (P1 → P2 → P3 → P4 → water) are marked with a black rectangle, whereas solutions appearing in the order P1 → P2 → P3 to P4 or P1 → P2 → P3 to water were marked with a light green rectangle.

As observed, water overbears aqueous L-fucose and L-proline making it hard to distinguish solutions of different concentrations from each other. However, some systematic differences between water and aqueous solutions were detected. The peaks of L-fucose solutions seem to have shifted away from the peak of the water between 1469–1662 nm, which is within the region where pure L-fucose differs from pure water. NIR absorption bands in this wavelength area belong to the 1st overtone vibrational OH stretching and water (Table 3) [62–64]. Although, differences between the aqueous solutions and water were observed (Figures 3 and 4) visually, a fine k-nearest neighbors (kNN) were used to investigate spectral differences between the L-fucose and L-proline solutions of different concentrations, as well as between the solutions and water. Optimal cross-validation accuracy (75.97%) was achieved with SNV normalized 1st derivative spectra (Savitzky–Golay filter: degree = 3, window = 99 for 319–1100 nm, and degree = 3, window = 17 for 943–2500 nm). A confusion matrix of a fine kNN classification (Figure 6) shows the number of observations in each cell's true and predicted classes. The F1-score for each class of a fine kNN classification is presented on the right side of the confusion matrix (Figure 6). The fine kNN classification shows that the NIRS measurements in this study are not as sensitive for the low concentration solutions (i.e., F3/P3 and F4/P4) as it is for higher concentrations.

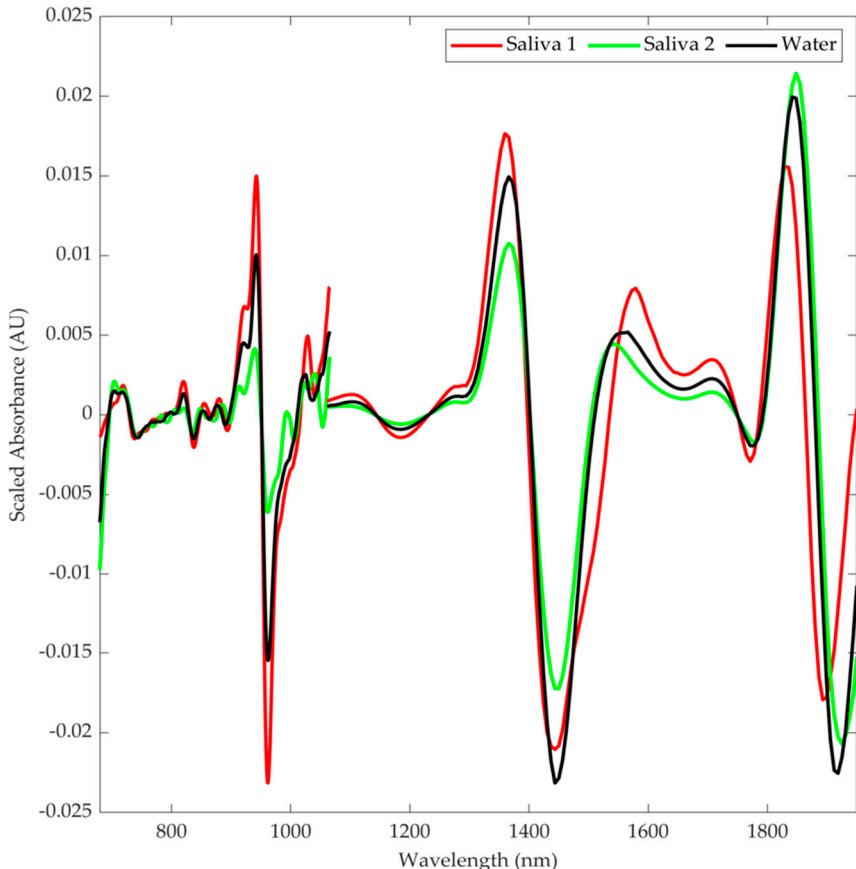

**Figure 5.** The 2nd derivative NIR spectra of two saliva samples of oral cancer patients and pure water as a comparison. Saliva 1 was collected from a 73-year-old male patient and saliva 2 was collected from a 76-year-old male patient.

Both saliva samples and water have peaks between 940–1068 nm, 1400–1553 nm, and 1840–1950 nm (Figure 5). These peaks were assigned to 2nd overtone OH stretching and water, 1st overtone OH stretching and water, and to combination OH stretching and water, respectively. Both L-proline and saliva sample 2 have two peaks between 730–1068 nm (Figures 1 and 4), whereas water has only one strong peak between 909–1068 nm (Figures 2 and 5). In this NIR spectral region appear 3rd overtone NH and CH stretching bands, water has an absorbance band and OH has 2nd overtone stretching band (Tables 3 and 4). The peak transition of saliva sample 1 was observed between 1530–1650 nm (Figure 5). That region has 1st overtone NH and OH stretching. Two peaks between 1476–1778 nm, that have higher scaled absorbance than water or saliva sample 2 were observed with saliva sample 1. In this region are 1st overtone OH stretching, water, and 1st overtone CH stretching bands [62].

*Reproducibility of the Sample Deposition Method*

The p-values of ANOVA for F1, F2, F3, F4, P1, P2, P3, and P4 solutions are 0.84, 0.34, 0.99, 0.99, 0.99, 0.98, 0.97, and 0.95, respectively, showing that there is not much variation between each measurement.

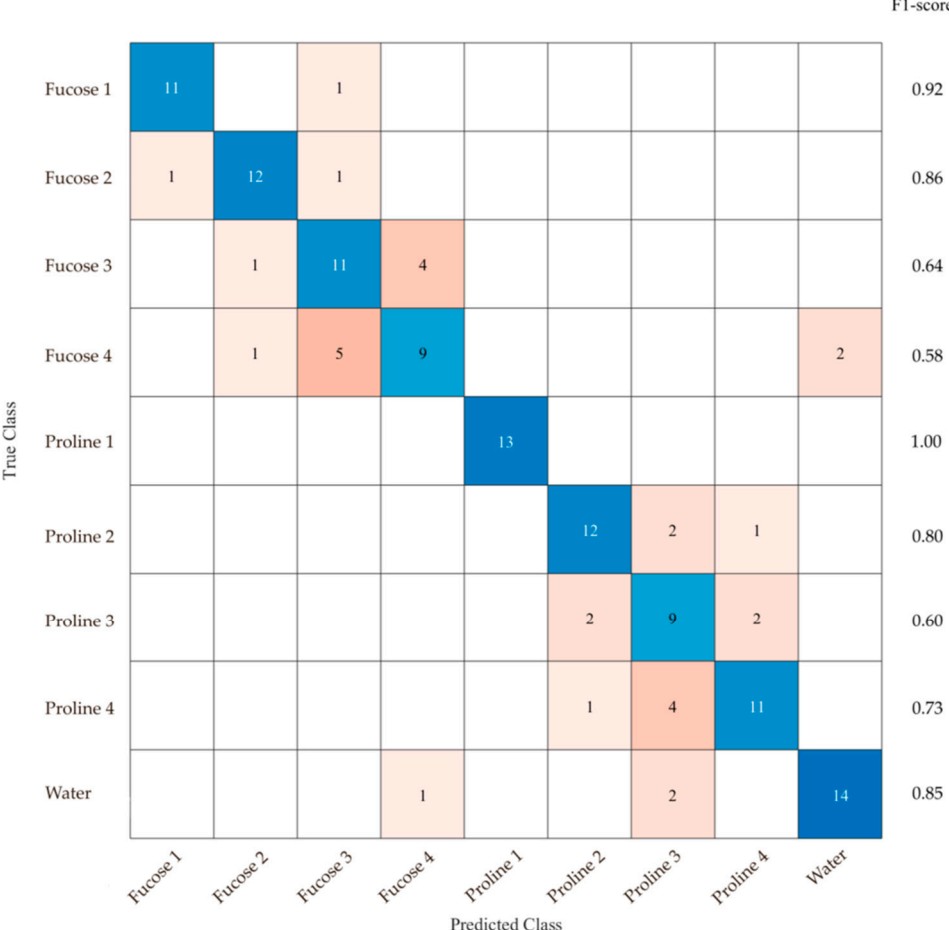

**Figure 6.** Confusion matrix of a cross-validated fine kNN classifier. F1-score of each class of a fine kNN classification are presented on the right side of the figure.

## 4. Discussion

This work aimed to study the ability of NIRS to measure and distinguish known oral cancer biomarkers from aqueous solutions. One hypothesis of the study was that solutions with different concentrations would be distinguishable from each other and water. The results indicate that NIRS can distinguish between the different concentrations of aqueous solutions using even a probe that is not specifically designed for measuring liquid samples. The spectra of pure solid L-fucose and L-proline (Figure 2) are noticeable different from that of water, but their diluted solutions and saliva (Figures 3–5) have more similar spectra with water. However, there are noticeable differences between solutions and water (Figures 3 and 4, and Tables 3 and 4) as noted in the present results. The differences between the spectra of L-fucose and L-proline solutions and water were also studied with a fine kNN for wavelengths between 680–1400 nm and 1550–1850 nm (Figure 6). The fine kNN, with validation accuracy of 75.97%, and the F1-scores (Figure 6) show that the sensitivity of the NIRS measurement is relatively poorer for solutions with lower concentrations. This might be due to the small number of measured spectra. The second hypothesis of this study was whether L-fucose and L-proline would be distinguishable on NIR absorbance spectra of two saliva samples of patients with diagnosed OSCC. This would indicate that the NIRS can recognize OSCC patients having higher concentrations of L-fucose and lower concentrations of L-proline from patients that do not suffer from OSCC as was the case in the NMR study by Mikkonen et al. [45]. Visual observation of the spectra shows that saliva sample 1 has two [45] absorbance peaks in spectral region 1476–1778 nm, and the peaks are stronger than those of water and saliva sample 2 (Figures 2 and 5). This finding is logical as sample 1 has an order of magnitude higher L-fucose concentration

(694 μM) than saliva sample 2 (67 μM). Within the same wavelength area, saliva sample 1 has a peak transition between 1530–1650 nm that is shifted away from the nearby peak of water (Figure 5). This might be due to the high L-fucose concentration of sample 1. Saliva sample 1 has a higher L-proline concentration (799 μM) than saliva sample 2 (157 μM) but interestingly, the saliva sample 2 has a stronger peak between 730–915 nm (Figure 5) and sample 1 has a stronger peak between 915–1068 nm in the spectral region of 730–1068 nm (Figure 5). To confirm these findings, the measurements should be repeated with a higher number of saliva samples collected from both OSCC patients and healthy controls.

Murayama et al. [53] developed a tube capillary method to study small volume liquid samples. In another study by Murayama et al. [38], authors utilized the method to measure FT-NIR spectra from saliva and distinguished oral cancer patients from healthy controls. In our study, spectral region 300–2500 nm was measured while the analysis was focused on the NIR region between 680–1950 nm. Murayama et al. [38] measured NIR spectra between 8000–5500 $cm^{-1}$ (1250–1818 nm). In their study, Murayama et al. [38], used only a single 2.5 μL drop of saliva, which is eight times less than in our study. Similar to our study, they found the water to dominate the spectra. In the study by Murayama et al. [38], broadband around 6900 $cm^{-1}$ (1449 nm) and peaks around 7072, 6839, and 5613 $cm^{-1}$ (1414, 1462, 1781 nm, respectively) were assigned to water. The first overtones of CH and NH stretching modes were observed within 6500–5500 $cm^{-1}$ (1538–1818 nm). Compared to Murayama et al. [38], our study requires less preparation as the sample can be pipetted straight on top of the reflective surface. Furthermore, our method is not dependent on circulating water as Murayama et al. [38] were. Our study shows that NIRS can distinguish different concentrations of aqueous biomarker solutions which can be further developed to distinguish patients with OPMD from patients suffering from oral cancer. Rai et al. [47] found using UV-VIS spectrophotometry that serum L-fucose levels are elevated in patients is suffering from oral cancer $13.85 \pm 4.34$ mg/dL ($0.1385 \pm 0.0434$ mg/mL) compared to patients suffering from oral leukoplakia $8.95 \pm 1.92$ mg/dL ($0.0895 \pm 0.0192$ mg/mL), and healthy controls $5.29 \pm 2.18$ mg/dL ($0.0529 \pm 0.0218$ mg/mL).

Rekha et al. [33] used NIR Raman spectroscopy to characterize salivary biomarkers to discriminate the oral premalignant and malignant conditions. They observed wavelength regions of molecular bonds of major components in the saliva between 800–1800 $cm^{-1}$ (5555–12,500 nm) and did a principal component analysis (PCA) together with linear discriminant analysis (LDA). Their sample set was larger and more divergent than in this study, but they concluded that more studies identifying the key biomolecules are required. Our preliminary study used two key salivary metabolomic biomarkers but further studies with larger sample sets from diverse patient groups are required. Mikkonen et al. [40] conducted a Fourier transformation infrared and photoacoustic (FTIR-PA) spectroscopy study using saliva collected from healthy patients whereas our two samples were from OSCC patients. They found three major spectral bands characterizing proteins 1500–1750 $cm^{-1}$ (5714–6666 nm), carbohydrates 1050–2000 $cm^{-1}$ (5000–9523 nm), and $SCN^-$ anions. They compared known $SCN^-$ concentrations to spectral data of saliva samples obtaining a correlation of $r > 0.990$ for transmission and $r = 0.967$ for PA mode. Similarly, further studies comparing L-fucose and L-proline solutions of known concentrations against NIR spectra of larger sets of saliva collected from healthy donors and patients suffering from oral malignancies could produce a statistically significant correlation.

One of the advantages of our study was the small sample volume required to measure a NIR absorbance spectrum. In the NMR spectroscopy study by Mikkonen et al. [45], the sample volume size was 0.5 mL. In our study, we found that one 20 μL drop of aqueous solution and saliva was enough to measure the NIR spectrum. This means that less saliva is required from the patient. Saliva drops dry non-homogenously on the silicon sample surface [41]. One drop of saliva spreads unevenly on the reflective material which is a disadvantage. One, three, and six 15 μL drops of saliva were dried on a silicon surface in an FTIR-PA spectroscopy study by Mikkonen et al. [40]. Another advantage of our study is the small sample preparation time required before measuring. Pipetting one drop of liquid

on a sample holder for measuring without having to dry the drop, as in our study, is less time-consuming than drying several drops onto a sample holder, which could take up to 20 h [40], before measuring. The NIRS set-up used in this study is mobile and clinically applicable; furthermore, a probe specifically designed for liquids can enhance the technique even further.

The NIR spectra of two saliva samples differ from water's spectrum in L-fucose and L-proline specific wavelength areas that were identified. A fine kNN validation was carried out, and it showed relatively poorer sensitivity for the solutions that had low concentrations. The results of this proof-of-concept study are qualitative as the sample set was small, and due to the slight uneven spreading of the sample drops. The volume of the drop should be 20 µL to avoid saturation of the signal. To get quantitative results, measurements with larger sets of saliva samples and a liquid measurement-specific probe are required. Saliva samples should be collected from patients suffering from OPMDs, from patients suffering from oral cancer, and from healthy controls. Different amounts of L-fucose and L-proline should be added to saliva to see if there are differences. These samples can then be used to make a multivariate model that could predict based on the concentrations of L-fucose and L-proline in saliva if the sample belongs to a patient suffering from for example oral leukoplakia or OSCC. This model could then be used in a clinical environment to help dentists and clinicians to detect oral lesions at an early stage and to decide whether the patient should be referred further to confirm the potential findings as malignant. The setup in this study could be used to study the condition of the oral mucosal tissue from the patient sitting in the dentist's chair. As the saliva is present on the surface of the mucosa, the method should be able to account for saliva as well. Our study provides the first step to support developing such a method. Compared to histopathologic assessment, screening for oral diseases using NIRS analysis of saliva is more comfortable for the patient.

## 5. Conclusions

NIRS can be used to detect differences in spectra of the samples with known concentrations of known oral cancer biomarkers (L-fucose and L-proline) in aqueous solutions. To the authors' best knowledge, no previous study has utilized NIRS to distinguish the spectral differences of various concentrations of oral cancer biomarkers from aqueous solutions or from saliva. The next step in research would be to spike saliva samples of healthy volunteers with different concentrations of L-fucose and L-proline and measure the NIRS spectra of those spiked samples. The sensitivity of the measurements could be enhanced using more samples and adapting a multivariate analysis to the larger data set. The measurement setup could be further enhanced by utilizing a probe dedicated to fluid measurements. This study provides a basis to further develop the investigational method to achieve quantitative and reproducible results and to develop NIRS setup to a feasible screening method to be able to carry out preclinical test series.

**Supplementary Materials:** The following are available online at https://www.mdpi.com/article/10.3390/app11209662/s1, Table S1: Data sheet.

**Author Contributions:** Conceptualization, S.M. and R.L.; methodology, M.O.H. and J.K.S.; software M.O.H. and J.K.S.; validation, M.O.H.; formal analysis, M.O.H. and J.K.S.; investigation, M.O.H. and J.K.S.; resources, S.M., W.A.G.-A., A.K. and R.L.; data curation, M.O.H., J.K.S. and S.M.; writing—original draft preparation, M.O.H.; writing—review and editing, M.O.H., J.K.S., S.M., W.A.G.-A., A.K., R.L.; visualization, M.O.H.; supervision, R.L.; project administration, S.M. and R.L.; funding acquisition, M.O.H., S.M., A.K., R.L. All authors have read and agreed to the published version of the manuscript.

**Funding:** This research was funded by Business Finland (former TEKES), grant number 52/31/2014. The M.O.H. was funded by Waldemar von Frenckells stiftelse.

**Institutional Review Board Statement:** The study was conducted according to the guidelines of the Declaration of Helsinki, and approved by the Ethics Committee for Human Studies of Piracicaba Dental School, Brazil (protocol code 142/2010).

**Informed Consent Statement:** Informed consent was obtained from all subjects involved in the study.

**Data Availability Statement:** The data presented in this study are available in this article or as Supplementary Materials.

**Acknowledgments:** We thank Jari Leskinen for his help in the preparation of the L-fucose and L-proline reference sample tablets.

**Conflicts of Interest:** The authors declare no conflict of interest. The funders had no role in the design of the study; in the collection, analyses, or interpretation of data; in the writing of the manuscript, or in the decision to publish the results.

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
