# Peer review of "Feasibility of Near-Infrared Spectroscopy for Identification of L-Fucose and L-Proline—Towards Detecting Cancer Biomarkers from Saliva"

_applsci, doi:10.3390/app11209662_

Round 1

Reviewer 1 Report

Comments have been attached in the file below

Reviewer 2 Report

In this manuscript the authors employ NIR to assess the feasibility of L- fucose and L-proline detection in two saliva samples from oral cancer patients. The study is well-designed and scientifically sound, however there are a few points to consider:

  1. The two samples employed were part of a previous study and this should be stated clearly in the Materials section.
  2. Given the available dataset (reference samples and saliva samples from patients), I would expect from the authors to assess the prediction of the biomarker concentration in the saliva samples (through a calibration curve for example) and then compare with the values measured with NMR in the Mikkonen et al. study. That would help to address the quantitative aspect of the study. Until then, it is misleading to state in the abstract that: ‘’ The preliminary study shows that the NIRS can be utilized to quantify biomarker concentrations in solutions.’’
  3. 2: please clarify that the spectra are recorded from ‘’reference samples’’ in the legend.
  4. Materials Section: ‘’ ..three independent aqueous solutions were prepared’’- it looks like they are four (F1-F4 and P1- P4)?
  5. Please be consistent with solution naming between text, Table 1, Fig. 3 and 4.
  6. 1 should: a) follow section 2.2 and b) is not easy to understand the different parts of the setup as panels overlap with parts of the photo. Please replace it with a clearer representation of the instrument.
